# Improving Zero-shot Reader by Reducing Distractions from Irrelevant Documents in Open-Domain Question Answering

**Sukmin Cho**     **Jeong yeon Seo**     **Soyeong Jeong**     **Jong C. Park***
School of Computing
Korea Advanced Institute of Science and Technology
{nelllpic,yena.seo,starsuzi,jongpark}@kaist.ac.kr

## Abstract

Large language models (LLMs) enable zero-shot approaches in open-domain question answering (ODQA), yet with limited advancements as the reader is compared to the retriever. This study aims at the feasibility of a zero-shot reader that addresses the challenges of computational cost and the need for labeled data. We find that LLMs are distracted due to irrelevant documents in the retrieved set and the overconfidence of the generated answers when they are exploited as zero-shot readers. To tackle these problems, we mitigate the impact of such documents via **D**istraction-aware **A**nswer **S**election (DAS) with a negation-based instruction and score adjustment for proper answer selection. Experimental results show that our approach successfully handles distraction across diverse scenarios, enhancing the performance of zero-shot readers. Furthermore, unlike supervised readers struggling with unseen data, zero-shot readers demonstrate outstanding transferability without any training.

## 1 Introduction

Open domain question answering (ODQA) is a task for answering questions with the evidence documents fetched from a large corpus (Voorhees and Tice, 2000). A *retrieve-read* framework has achieved remarkable performance in ODQA by fine-tuning the language models with labeled datasets (Lee et al., 2019; Karpukhin et al., 2020; Izacard and Grave, 2021). The emergence of large language models (LLMs) has enabled the exploration of zero-shot approaches in this framework, with less emphasis on the reader component (Sachan et al., 2022; Chuang et al., 2023; Levine et al., 2022).

Utilizing an LLM as a reader provides an advantage in the generalization ability with the rich world knowledge, unlike conventional small-sized supervised readers (Karpukhin et al., 2020; Izacard

---

*Corresponding author

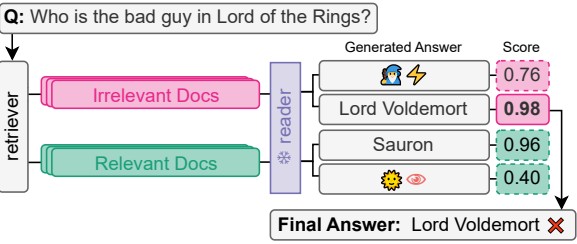

Figure 1: An overview of distraction from the irrelevant documents when exploiting LLM as a zero-shot reader.

and Grave, 2021). While the supervised readers show remarkable performance on ODQA, they are hampered by two weaknesses: the high computational cost involved in training and the necessity of annotated query-document datasets. These limitations impede the transferability of readers to diverse tasks and domains. To solve this, we aim to validate the feasibility of using an LLM as a reader, leveraging its inherent advantages while reducing the aforementioned limitations.

However, the performance of an LLM in various tasks is easily distracted by irrelevant documents (Li et al., 2022; Shi et al., 2023), underscoring the importance of resolving these challenges in ODQA. The tendency of an LLM to generate incorrect answers becomes apparent when reading retrieved sets that include irrelevant documents. These documents, while related to the query, may lack the necessary information to provide an answer, leading to the occurrence of hallucination. This emphasizes the need for proper handling of such documents to fully harness the potential of an LLM, thereby achieving reliable performance as a reader. This paper addresses the requisite of hallucination mitigation to validate the possibility of an LLM as a zero-shot reader.

In this paper, we propose **D**istraction-aware **A**nswer **S**election (DAS), handling the challenges posed by irrelevant documents and overconfident scores as shown in Figure 1. First, we provide

models with an "unanswerable" instruction, allowing them to abstain from answering. Then, we adjust the answer scores by reflecting the query generation score as the relevance between the given query-document pairs. These approaches reduce the impact of irrelevant documents and improve the selection of the correct answer from the relevant document.

We evaluate our proposed method on representative ODQA benchmarks with two publicly open LLMs, FLAN-T5 (Chuang et al., 2023) and OPT-IML-MAX (Iyer et al., 2022). This results in substantial performance improvements achieved by ours compared to a naïve LLM across all scenarios. Note that ours effectively alleviates the hallucination induced by irrelevant documents by enhancing the robustness against the number of documents that are read. Furthermore, an LLM with our method exhibits excellent transferability compared to the supervised reader, offering the untapped potential of an LLM as a zero-shot reader.

Our contributions in this paper are threefold:

- We tackle the distraction incurred by irrelevant documents and overconfident scores when exploiting an LLM as a zero-shot reader in ODQA tasks.

- We introduce **D**istraction-aware **A**nswer **S**election (DAS) for a zero-shot reader, with the unanswerable instruction and the score adjustment eliciting its deductive ability.

- We empirically verify the efficacy of our proposed approach in effectively mitigating hallucination and unlocking the feasibility of zero-shot readers with a generalization ability.

## 2 Related Work

**Zero-shot Approach in ODQA**   The advent of an LLM has shown the potential that it can be used in two stages without parameter updates. For the retrieval stage, an LLM is exploited as a re-ranker via query generation or document permutation (Sachan et al., 2022; Cho et al., 2023; Sun et al., 2023) or expanded query to diverse pseudo queries for improving the performance of supervised retrievers (Liu et al., 2022; Yu et al., 2023; Chuang et al., 2023). For the reader stage, Levine et al. (2022) attempted to utilize an LLM as a zero-shot reader, addressing the irrelevant documents through a re-ranker. In this study, we focus on a fully zero-shot reader without an additional module.

**Distraction from Noisy Input**   Recent work addresses the negative impact of noisy inputs when exploiting an LLM in diverse tasks. LLMs are easily distracted by the noisy input having incorrect or irrelevant information on machine reading comprehension tasks (Li et al., 2022; Su et al., 2022; Shi et al., 2023). However, the ODQA task increases the complexity, where large-scale document sets appear within unrelated documents. Given the impact of distracting sentences in QA  (Khashabi et al., 2017; Jia and Liang, 2017; Ni et al., 2019), our approach aims to alleviate them.

## 3 Method

### 3.1 Preliminaries

To adopt the LLM into the reader, we define a two-step answering pipeline consisting of answer candidate generation and final answer selection.

**Answer Candidate Generation**   The LLM $M$ generates answer candidate $a_i$ based on the given query $q$, the evidence document $d_i$ in retrieved documents $D$ and the reading comprehension instruction $\rho_{rc}$ via greedy decoding. This process results in an answer candidate set $S = \{(a_i, d_i)\}_{i=1}^k$.

**Final Answer Selection**   We select the final document-answer pair $p^* = (a^*, d^*)$ from an answer candidate set $S$ based on the generation probability $P_M(a_i|q, d_i, \rho_{rc})$ as the answer score. The document-answer pair with the highest probability is chosen as the most likely correct answer.

### 3.2 Problem Definition

We address selecting the incorrect answer as the final one as caused by distraction from the irrelevant documents $d_N$. The irrelevant documents present a challenge as they cannot be used to infer the correct answer, misleading the LLM to generate incorrect but plausible answers $a_N$. The presence of such answers $a_N$ in the answer set $A$ can result in obstacles during the final answer selection.

Another challenge arises from the overconfident scores, making it difficult to discern the incorrect answers $a_N$ from the documents $d_N$. The LLM, being an auto-regressive model, tends to produce text sequences with high probabilities when using greedy decoding. Consequently, it becomes hard to accurately determine the correct answer $a^*$ based on the generation probabilities, especially when it also includes incorrect answers like $a_N$.

| Retriever | Reader | Top-20 | | | | Top-100 | | | |
|---|---|---|---|---|---|---|---|---|---|
| | | NQ | TQA | WebQ | SQD | NQ | TQA | WebQ | SQD |
| **BM25** | **FLAN-T5-XL** | 23.37 | 52.68 | 16.19 | 19.40 | 17.86 | 46.12 | 15.83 | 15.79 |
| | w/ DAS | 31.51 | **64.54** | 20.14 | **39.39** | 33.84 | **68.86** | **25.90** | **46.71** |
| | | (+40.8%) | (+22.5%) | (+24.4%) | (+103%) | (+89.5%) | (+49.3%) | (+63.6%) | (+195%) |
| | **OPT-IML-MAX** | 20.21 | 53.21 | 23.38 | 22.93 | 16.32 | 46.57 | 18.71 | 18.34 |
| | w/ DAS | 28.72 | 56.95 | 24.10 | 32.37 | 29.76 | 59.87 | 24.10 | 37.74 |
| | | (+42.1%) | (+7.0%) | (+3.1%) | (+41.2%) | (+82.4%) | (+28.6%) | (+28.8%) | (+105%) |
| **DPR** | **FLAN-T5-XL** | 22.43 | 47.44 | 20.50 | 12.85 | 15.90 | 39.17 | 16.55 | 10.30 |
| | w/ DAS | **37.77** | 64.48 | **26.98** | 26.66 | **37.96** | 68.22 | 25.18 | 34.12 |
| | | (+68.4%) | (+35.9%) | (+31.5%) | (+107%) | (+138%) | (+74.2%) | (+52.1%) | (+231%) |
| | **OPT-IML-MAX** | 23.28 | 50.24 | 21.58 | 16.03 | 16.65 | 43.67 | 19.42 | 14.47 |
| | w/ DAS | 33.69 | 56.61 | **26.98** | 21.97 | 32.95 | 59.05 | 25.54 | 28.46 |
| | | (+44.7%) | (+12.7%) | (+25.0%) | (+37.1%) | (+97.9%) | (+35.2%) | (+31.5%) | (+96.7%) |

Table 1: EM accuracy of the final answer among the answer candidates generated from the top-$k$ retrieved documents. The best scores are marked in **bold**. The number in parentheses means the improvement percentage from DAS.

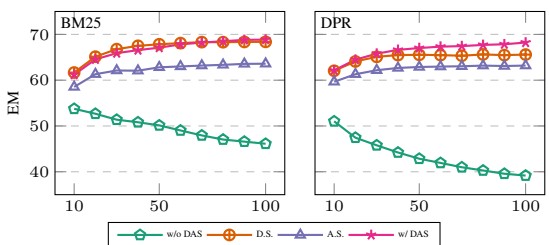

Figure 2: EM accuracy depending on the number of the documents retrieved by BM25 and DPR on TQA.

| Reader Model | Train Set | NQ | TQA | SQD | RQA |
|---|---|---|---|---|---|
| DPR† | Multi | 41.5 | 56.8 | 29.8 | - |
| FiD-base | NQ | 45.1 | 54.1 | 34.1 | 29.8 |
| | TQA | 26.9 | 64.5 | 27.5 | 33.2 |
| FiD-large | NQ | 50.8 | 59.2 | 36.2 | 34.0 |
| | TQA | 30.9 | 69.0 | 31.5 | 34.4 |
| FLAN-T5-XL w/ DAS | - | 34.0 | 57.2 | 43.5 | 35.8 |

Table 2: Comparison of ours against the supervised readers on the test set under the condition of exploiting DPR. † denotes the performance from its paper.

### 3.3 Distraction-aware Answer Selection

We present simple yet effective **D**istraction-aware **A**nswer **S**election (DAS) for a zero-shot reader. We aim to reduce the negative impact of irrelevant documents in a two-step answering pipeline. Initially, we offer an option to refuse responses to irrelevant documents via an unanswerable instruction. To improve the final answer selection, we incorporate the relevance of the query-document pair into the scoring process.

**Document Selection (D.S.)** We utilize the unanswerable instruction to enhance the deduction capability by giving the option not to respond. We exclude responses that belong to the unanswerable response set $U$ as follows:

$$S' = \{(a_i, d_i) | a_i \notin U, (a_i, d_i) \in S\} \quad (1)$$

We construct an unanswerable response set $U = \{$"Unanswerable", "Answer not in context"$\}$. The answers in $U$ are judged unanswerable as if the reader rejects to respond to the irrelevant documents.

**Answer Selection (A.S.)** Then, we adjust the answer score by multiplying the query generation score in consideration for the query-document relevance. This is formulated as follows:

$$(a^*, d^*) = \underset{(a_i', d_i') \in S'}{\arg\max} \; P_M(a_i'|q, d_i', \rho_{rc}) \cdot P_M(q|d_i', \rho_{qg}) \quad (2)$$

where $\rho_{qg}$ denotes the query generation instruction.

The query generation score from the given document is computed as:

$$\log P(q|d) = \frac{1}{|q|} \sum_t \log P(q_t|q_{<t}, d) \quad (3)$$

## 4 Experimental Setup

**Dataset** We experiment on **Natural Question** (NQ) (Kwiatkowski et al., 2019), **TriviaQA** (TQA) (Joshi et al., 2017), **WebQuestions** (WebQ) (Berant et al., 2013) and **SQuAD** (Rajpurkar et al., 2016) (SQD). [1] For annotated evidence documents for query, the development sets of each dataset are used.

**Retriever** We employ the representative sparse retriever, **BM25** (Robertson and Zaragoza, 2009), and the dense one, **DPR** (Karpukhin et al., 2020).

---

[1]Following the settings from Karpukhin et al. (2020), the English Wikipedia dump from Dec 20, 2018, is used.

| Reader | Correct Answer | Incorrect Answer | Total Answer |
|---|---|---|---|
| **FLAN-T5-XL** | 5.50 (5.50%) | 94.50 (94.50%) | 100 |
| w/ DAS. | 2.89 (21.93%) | 10.27 (78.07%) | 13.16 |
| **OPT-IML-MAX** | 5.13 (5.13%) | 94.87 (94.87%) | 100 |
| w/ DAS. | 2.85 (10.97%) | 23.14 (89.03%) | 26.00 |

Table 3: Average number of answers in the candidate set $S$. The number in parentheses represents the proportion relative to the total number in $S$.

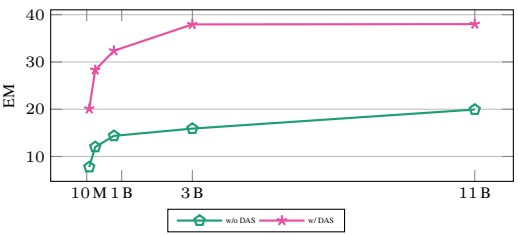

Figure 3: EM accuracy depending on the model size. The exploited models are the families of FLAN-T5.

**Language Model** We select two publicly open LLMs: **1) FLAN-T5** (Chung et al., 2022) is the family of T5 (Raffel et al., 2020) with instruction tuning; **2) OPT-IML** (Iyer et al., 2022) is the fine-tuned version of OPT (Zhang et al., 2022) by instruction meta learning. We exploit FLAN-T5-XL containing 3B parameters and OPT-IML-MAX-1.3B in our main experiments.

**Metrics** In our evaluation, we employ the exact match (EM) accuracy metric to assess whether the reader generates the same answer as the annotated answer, after applying normalization techniques such as punctuation removal. We adhere to the same normalization process utilized in previous works (Chen et al., 2017; Lee et al., 2019).

**Implementation Details** The reading comprehension instruction is "*Read the following context and answer the question*". We add "*If you don't know the answer, return unanswerable*" for the unanswerable instruction, as mentioned in Sanh et al. (2022). Also, we compute the query generation score, following settings from Sachan et al. (2022). More details are in Appendix B.

## 5 Result

### 5.1 Main Result

Table 1 demonstrates the significant performance improvements achieved by DAS regardless of retrievers, LLMs, and datasets. Our method achieves an increase in EM of 64% on average against the default, with a remarkable improvement of 231%.

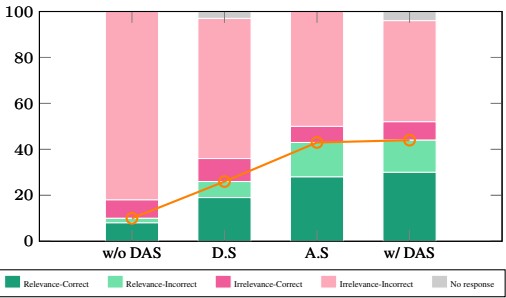

Figure 4: Distribution of answer pairs $p^*$ based on document-query relevance and answer correctness.

As the size of the retrieved set increases the likelihood of including relevant documents, the reader should be robust to irrelevant documents. Nevertheless, the presence of disctraction becomes apparent as indicated by the performance decline without DAS, as shown in Table 1 and Figure 2, when processing more documents. This challenge is addressed by mitigating the negative impact of irrelevant documents. Our approach achieves an average enhancement of 17% in EM when reading 100 documents compared to 20. This shows the robustness of our approach in handling the problem stemming from the irrelevant documents.

Also, we find that when reading 100 documents, the use of documents collected through BM25 has a more positive impact on the performance of the reader compared to documents from DPR. This finding is noteworthy, especially considering that DPR generally performs better in retriever tasks. When employing a zero-shot reader, it cannot be definitively concluded that improved performance of the retriever will necessarily lead to enhanced reader performance. More details are in Appendix C.

**Comparison against Supervised Reader** We directly compare with the supervised readers on the aforementioned datasets and an additional held-out dataset, RealTimeQA (RQA) (Kasai et al., 2022). As shown in Table 2, the zero-shot reader with ours shows robust performance compared to supervised readers, DPR (Karpukhin et al., 2020) and FiD (Izacard and Grave, 2021), which perform poorly on unseen data such as SQuAD and RQA. We highlight their potential as a valuable alternative that avoids the limitations and costs associated with supervised readers.

## 5.2 Analysis

Our analysis is conducted on NQ with the top 100 documents retrieved by DPR with FLAN-T5-XL. Detailed analysis are in Appendix D.

**Impact of Model Size** We conduct experiments to assess the impact of model size on performance. As shown in Figure 3, the results demonstrate that even with smaller models, ours maximizes the performance of an LLM as a zero-shot reader. This indicates that our approach enables LLMs to function effectively as zero-shot readers, even without the need for extensively large parameter sizes.

**Answer Candidate Set** We examine the effects of applying DAS on the answer candidate set $S$ as presented in Table 3. Our findings highlight a remarkable shift in the distribution of answers, with changes of 16.43%p and 5.84%p observed in each reader. Substantial increases in the ratio of correct answers demonstrate that ours effectively mitigates the inclusion of incorrect answers from irrelevant documents.

**Final Answer Pair** Figure 4 illustrates an analysis of the distribution of the final answer pair $p^*$. The results provide evidence that ours successfully selects documents that are relevant to the given query and enable the extraction of a higher number of correct answers from the relevant documents. Additionally, ours shows a reduction of approximately 5% in the rate of incorrect answers generated from irrelevant documents.

## 6 Conclusion

In this paper, we propose **D**istraction-aware **A**nswer **S**election (DAS) to address the irrelevant documents in the retrieved set when an LLM is used as a zero-shot reader. To validate its capability, we define hallucination caused by irrelevant documents and overconfident answer scores in ODQA setting. Ours aims to mitigate the impact of these aspects by incorporating unanswerable instruction and adjusting answer scores for better answer selection. Experimental results demonstrate the effectiveness of our proposal in handling hallucination across various scenarios, thereby improving the performance of ODQA benchmarks. Our approach, utilizing an LLM, showcases strong generalization capabilities across diverse datasets, distinguishing it from supervised readers and highlighting the potential of a zero-shot reader.

## Limitations

Our methodology utilizes a two-step pipeline to enhance the performance of an LLM as a zero-shot reader, addressing hallucination issues and leveraging its functionality. While ours fully elicit the inherent ability of the zero-shot reader from LLM, its effectiveness is dependent on the capabilities and characteristics of the LLM. For example, the prompt sensitivity of an LLM is one of the important aspects to consider, as different prompts may lead to varying results. Also, the performance of an LLM is size-dependent. Although our experiments have yielded consistent results in numerous cases, further investigation is required to evaluate our approach with larger LLMs. Despite these limitations, the zero-shot approach holds great promise in terms of cost-effectiveness and leveraging abundant world knowledge. As future advancements in LLMs are anticipated, we expect even greater improvements in performance over the state-of-the-art supervised readers.

## Ethics Statement

We acknowledge the possibility of bias or offensive answer sets in utilizing an LLM as a zero-shot reader. Since this paper primarily focuses on the mitigating impact of irrelevant documents in ODQA without parametric updates, addressing the issue of bias and offensive language within an LLM is beyond the scope of our paper. We are aware that ongoing research and efforts are being made by researchers to address these concerns and improve the ethical aspects of LLMs. It is expected that future advancements and research in the field will contribute to addressing these biases and ensuring an ethical use of LLMs.

## Acknowledgements

This work was supported by an Institute for Information and communications Technology Promotion (IITP) grant funded by the Korea government (No. 2018-0-00582, Prediction and augmentation of the credibility distribution via linguistic analysis and automated evidence document collection). This work was also supported by the Artificial intelligence industrial convergence cluster development project funded by the Ministry of Science and ICT (MSIT, Korea) & Gwangju Metropolitan City.

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

# A   Related Work

We describe the related work on unanswerable instruction and query generation score in our proposed method, **D**istraction-aware **A**nswer **S**election (DAS).

## A.1   Unanswerable Instruction

The unanswerable queries were introduced to ensure the effective discernment of query-document relevance (Rajpurkar et al., 2018). This approach was incorporated in the pre-training of LLMs when models cannot find the answer within the provided document (Wei et al., 2022; Sanh et al., 2022; Iyer et al., 2022). We revisit these approaches in a zero-shot setting to confirm the feasibility of the unanswerable instruction for filtering out irrelevant documents in the retrieved set.

## A.2   Document Ranking with Query Generation Score

The query generation score is a widely used measure of query-document relevance when ranking the documents (Nogueira dos Santos et al., 2020; Ju et al., 2021). Recently, LLMs serve as zero-shot re-rankers with outstanding performance gain by computing the measure (Sachan et al., 2022; Cho et al., 2023). To this end, we highlight the capacity of LLMs to ascertain the relevance between the query-document pair when exploiting them as a zero-shot reader.

# B   Experimental Setup

## B.1   Dataset

In our main experiments, we utilize a development set of four representative ODQA datasets for employing annotated evidence documents to analyze the impact of query-document relevance. We apply a filtering process to exclude some data that do not contain evidence documents. For a fair comparison against the supervised readers, DPR (Karpukhin et al., 2020) and FiD (Izacard and Grave, 2021) [2], we use the test set of each dataset which has already been preprocessed in Sachan et al. (2022) [3].

**Natural Question (NQ)**     (Kwiatkowski et al., 2019) was specifically crafted for ODQA tasks. It comprises queries from Google search engines and

the answers extracted from Wikipedia documents. In our experiment, a development set and a test set of NQ contain 6,515 and 3,610 queries, respectively.

**TriviaQA (TQA)**     (Joshi et al., 2017) was for reading comprehension dataset consisting of question-answer-envidence triplets. The queries are fetched from the quiz websites and the corresponding evidence documents are collected from the Wikipedia documents via the Bing search engine. In our experiment, a development set and a test set of TQA contain 6,760 and 11,313 queries, respectively.

**WebQuestions (WebQ)**     (Berant et al., 2013) collected the queries from Google Suggest API and its answer from the entities in Freebase. The evidence documents were defined as the highest-ranked documents from BM25 having the answer (Lee et al., 2019). We use a development set of WebQ consisting of 361 questions.

**SQuAD (SQD)**     (Rajpurkar et al., 2016) was based on manually annotated queries from Wikipedia documents. While SQuAD wasn't designed for ODQA tasks, it was widely used for evaluating reader performance. A development set and a test of SQuAD contain 8,886 and 10,570 queries, respectively.

## B.2   Instruction & Template

As LLMs are sensitive to instruction and templates when adopting the downstream tasks without parameter updates, we carefully select via iterative validation. The reading comprehension instruction is "*Read the following context and answer the question*" and the unanswerable instruction is "*Read the following context and answer the question. If you don't know the answer, return unanswerable*". When transmitting a query, a document, and an instruction to LLMs, we use the input template following the setting from Chung et al. (2022) and Iyer et al. (2022). The input templates are "{I}\n\nContext: {D}\nQuestion: {Q}" for FLAN-T5 and "{I}\n\nContext: {D}\nQuestion: {Q}\nAnswer: " for OPT-IML-MAX where I,D and Q denotes an instruction, a document, and a question, respectively.

---

[2]We evaluate FiD with the model checkpoints from their publicly opened repository.

[3]https://github.com/DevSinghSachan/unsupervised-passage-reranking

| Retriever | Reader | Top-10 | | | | Top-20 | | | | Top-50 | | | | Top-100 | | | |
|---|---|---|---|---|---|---|---|---|---|---|---|---|---|---|---|---|---|
| | | NQ | TQA | WebQ | SQD | NQ | TQA | WebQ | SQD | NQ | TQA | WebQ | SQD | NQ | TQA | WebQ | SQD |
| BM25 | FLAN-T5-XL | 23.4 | 53.8 | 15.1 | 20.8 | 22.4 | 52.7 | 16.2 | 19.4 | 19.5 | 50.1 | 18.4 | 17.1 | 17.9 | 46.1 | 15.8 | 15.8 |
| | w/ DAS | 28.5 | 61.2 | 21.6 | 35.8 | 31.5 | 64.5 | 20.1 | 39.4 | 32.8 | 67.1 | 24.1 | 43.7 | 33.8 | 68.9 | 26.0 | 46.7 |
| | OPT-IML-MAX | 20.8 | 54.1 | 23.0 | 23.7 | 20.2 | 53.2 | 23.4 | 22.9 | 17.6 | 50.2 | 20.1 | 20.4 | 16.3 | 46.6 | 18.7 | 18.3 |
| | w/ DAS | 26.8 | 54.5 | 22.7 | 29.7 | 28.7 | 57.0 | 24.1 | 32.4 | 29.6 | 59.1 | 27.3 | 35.7 | 29.8 | 59.9 | 24.1 | 37.7 |
| DPR | FLAN-T5-XL | 25.8 | 51.0 | 19.8 | 13.4 | 22.4 | 47.4 | 20.5 | 12.9 | 18.7 | 42.9 | 19.4 | 11.2 | 15.9 | 39.2 | 16.6 | 10.3 |
| | w/ DAS | 37.2 | 62.2 | 26.0 | 22.6 | 37.8 | 64.5 | 27.0 | 26.7 | 37.9 | 67.0 | 25.5 | 31.1 | 38.0 | 68.2 | 25.2 | 34.1 |
| | OPT-IML-MAX | 26.1 | 52.0 | 23.7 | 15.8 | 23.3 | 50.2 | 21.6 | 16.0 | 19.1 | 46.7 | 23.0 | 15.5 | 16.6 | 43.7 | 19.4 | 14.5 |
| | w/ DAS | 33.5 | 54.8 | 28.1 | 19.5 | 33.7 | 56.6 | 27.0 | 22.0 | 33.4 | 58.3 | 27.0 | 25.9 | 33.0 | 59.1 | 25.5 | 28.5 |

Table 4: Exact match accuracy of the final answer among the generated answers from top-$k$ retrieved documents for the open-domain question answering benchmarks.

| Reader | Relevant Document | | | Irrelevant Document | | |
|---|---|---|---|---|---|---|
| | Cor. | Inc. | NR. | Cor. | Inc. | NR. |
| NQ-Dev | | | | | | |
| FLAN-T5-XL | 1.58 | 1.09 | 0.01 | 3.91 | 91.29 | 2.12 |
| w/ DAS | 1.27 | 0.59 | 0.82 | 1.61 | 9.68 | 86.02 |
| OPT-IML-MAX | 1.51 | 1.16 | 0.01 | 3.62 | 86.33 | 7.36 |
| w/ DAS | 1.22 | 0.75 | 0.71 | 1.63 | 22.39 | 73.29 |
| TQA-Dev | | | | | | |
| FLAN-T5-XL | 3.61 | 1.54 | 0.01 | 13.08 | 78.86 | 3.02 |
| w/ DAS | 3.23 | 0.49 | 1.42 | 6.61 | 4.47 | 86.91 |
| OPT-IML-MAX | 3.79 | 1.32 | 0.02 | 13.16 | 75.62 | 6.29 |
| w/ DAS | 2.83 | 0.78 | 1.54 | 5.49 | 13.10 | 79.17 |

Table 5: Average number of answers in the answer candidate set $S$ including unanswerable response set $U$. Cor and Inc denote correct and incorrect answer, respectively. NR means no response to the documents.

## B.3 Environment

We conduct all experiments on A100 80GB GPUs. We use BEIR (Thakur et al., 2021) framework[4] for the retriever, BM25 and DPR. We employ FLAN-T5 and OPT-IML-MAX with 3B and 1.3B parameters publicly open on the Huggingface model hub[5] (Wolf et al., 2020).

## C Detailed Results

We provide more comprehensive results in terms of both top-10 and top-50 documents, as illustrated in Table 4 and Figure 6. In the absence of our proposed methodology, there is a noticeable decline in performance as the number of documents read increases. However, when employing DAS, we observe a reduction in the impact of hard-negative documents within the document set, resulting in an enhanced reader capability. DAS effectively mitigates the adverse effects of such documents and maximizes the overall performance of a reader.

In an ablation study, Figure 6 showcases the influence of document selection (D.S.) and answer selection (A.S.) within our proposed method. Both

[4] http://beir.ai/
[5] https://huggingface.co/models

selections contribute positively to enhancing the performance of LLM. However, in the case of OPT-IML-MAX, the impact of document selection is found to be insignificant. This observation suggests that OPT-IML-MAX, despite its ability to distinguish irrelevant documents based on instructions, falls short compared to FLAN-T5 in effectively addressing the hallucination.

## D Analysis

### D.1 Aanlaysis of Unanswerables

As shown in Table 5, we conduct an analysis of the model's responses to documents, including those that are excluded from the answer candidate set $S$ during the document selection process. While our method successfully reduced the number of responses from irrelevant documents, we observed a slight decrease in relevant documents. However, the primary focus of our methodology is on increasing the portion of correct answers by minimizing the number of incorrect answers originating from irrelevant documents. This aspect is key to our approach and contributes to the overall improvement of reader performance.

### D.2 Analysis of Overconfident Score

We conducted a verification to determine whether the answer score was indeed overconfident. As depicted in Figure 5, when DAS is not utilized, the incorrect answer exhibits a remarkably high generation probability, making it indistinguishable from the correct answer. However, upon implementing DAS, the scores are normalized, resulting in a discernible distribution disparity between correct and incorrect answers.

### D.3 Case Study

We present two curated examples in Table 6 to illustrate the effectiveness of our proposed approach in mitigating hallucination compared to naïve LLMs.

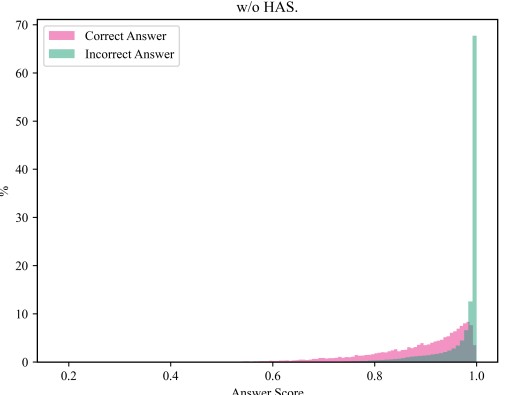 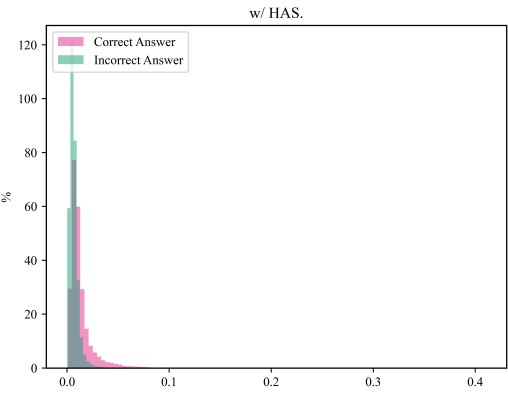

Figure 5: The analysis of answer score. The left plot is for FLAN-T5-XL without HAS and the Right plot is with HAS. Both experiments are on NQ development set with evidence documents retrieved by DPR.

In these examples, the naïve LLMs erroneously provide the answer "Straits of Mackinac" in unrelated contexts to "Lake Michigan-Huron" when given the query about "The Great Lakes". However, by employing our method, the correct answers are extracted from the relevant documents. This highlights the ability of our approach to alleviate hallucination and facilitate the accurate selection of appropriate answers based on contextual information.

Additionally, we showcase two error cases in Table 6. In these cases, the reader generates the correct answer based on the relevant document, but our approach produces plausible alternative answers. For instance, in response to the question "What is the deepest depth in the oceans?", the reader correctly identifies "Challenger Deep" based on another relevant document not included in annotated evidence set. While this answer is technically incorrect according to EM evaluation, it is difficult to perceive it as entirely incorrect when assessed qualitatively.

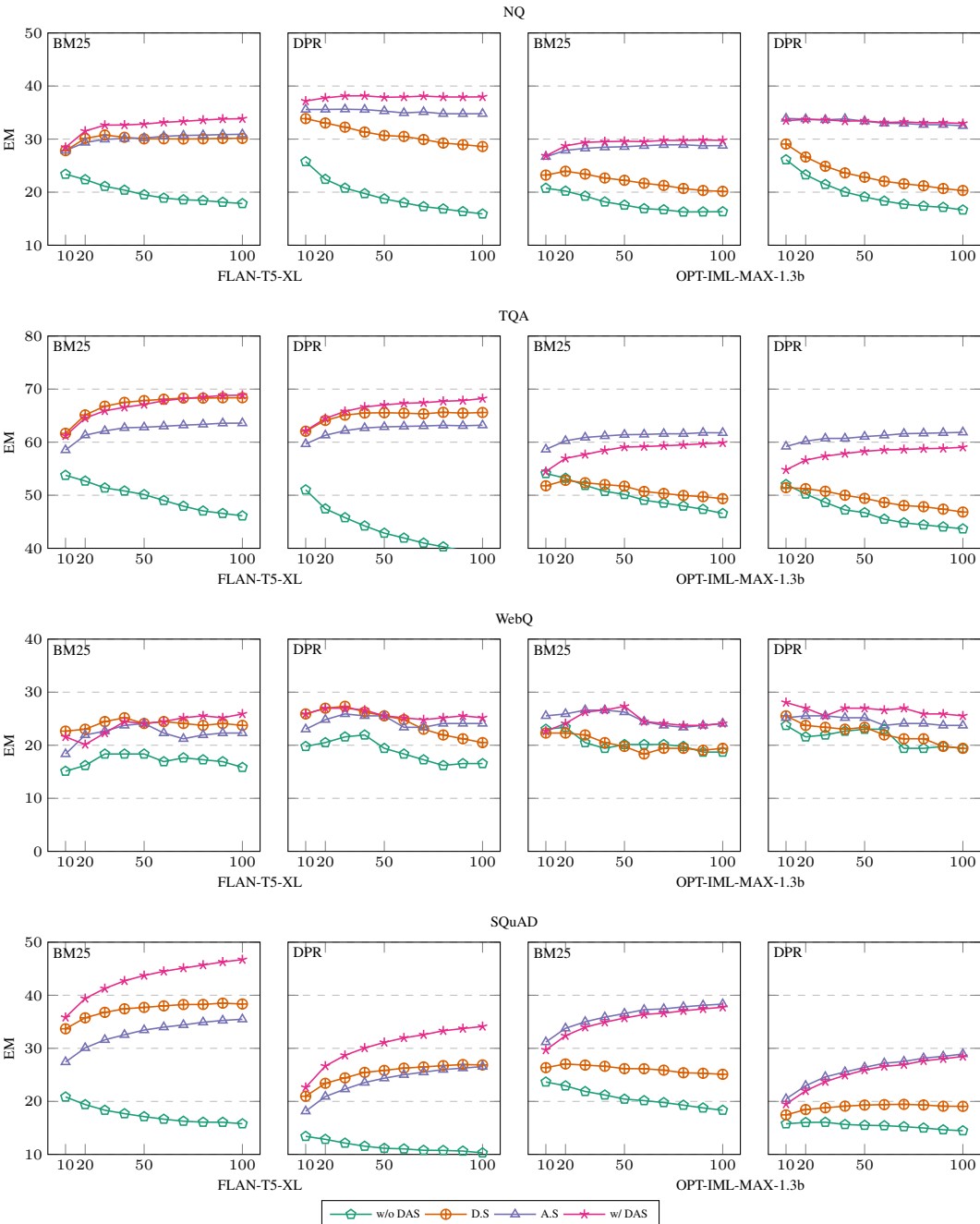

Figure 6: EM accuracy depending on the number of the retrieved documents.

|  | Case 1 | Case 2 |
|---|---|---|
| **Query** | Where do the great lakes meet the oceans? | Who was the creator of Victoria's Secret? |
| **Gold Answer** | the Saint Lawrence River | Roy Raymond |

| w/o DAS | | |
|---|---|---|
| **Final Document** | Lake Michigan–Huron, because they are one hydrological body of water connected by the Straits of Mackinac. The straits are wide and deep; the water levels (...) | Traci Paige Johnson is an American animator, television producer, and voice actress, most known for creating the Nick Jr. preschool television series, (...) |
| **Final Answer** | Straits of Mackinac | Traci Paige Johnson |

| w/ DAS | | |
|---|---|---|
| **Final Document** | The Great Lakes are a series of interconnected freshwater lakes located primarily in the upper mid-east region of North America, on the Canada–United States border, which connect to the Atlantic Ocean through the Saint Lawrence River. | Victoria's Secret is an American designer, manufacturer, and marketer of women's lingerie, womenswear, and beauty products. (...) Victoria's Secret was founded by Roy Raymond, and his wife Gaye Raymond, in San Francisco, California, (...) |
| **Final Answer** | the Saint Lawrence River | Roy Raymond |

|  | Error Case 1 | ErrorCase 2 |
|---|---|---|
| **Query** | Who plays mrs. potato head in toy story? | What is the deepest depth in the oceans? |
| **Gold Answer** | Estelle Harris | Mariana Trench |

| w/o DAS | | |
|---|---|---|
| **Final Document** | (...) After Mr. Potato Head saves three Pizza Planet Aliens (Jeff Pidgeon) from falling out of a Pizza Planet truck, his wife, Mrs. Potato Head (Estelle Harris) adopts them, making her husband upset. (...) | In the Challenger Deep, he and Lt. Don Walsh of the United States Navy were the first people to explore the deepest part of the world's ocean, and the deepest location on the surface of Earth's crust, the Mariana Trench, located in the western North Pacific Ocean. (...) |
| **Final Answer** | Estelle Harris | Mariana Trench |

| w/ DAS | | |
|---|---|---|
| **Final Document** | Pop singer Melanie Martinez released a song called "Mrs. Potato Head" on her debut album "Cry Baby". Mr. Potato Head is also in the Disney/Pixar "Toy Story films" voiced by Don Rickles. (...) | The Challenger Deep, located just outside the Trench Unit, is the deepest point in the Earth's oceans, deeper than the height of Mount Everest above sea level. (...) |
| **Final Answer** | Don Rickles | Challenger Deep |

Table 6: Examples of hallucination alleviation and error cases. FLAN-T5-XL is exploited as a reader on the Natural Question dataset.