# OpenReview forum: "Improving Zero-shot Reader by Reducing Distractions from Irrelevant Documents in Open-Domain Question Answering"
_EMNLP/2023/Conference — EMNLP 2023 Findings_

### Official Review · Reviewer_Y5qm · 2023-07-30

**Soundness:** 4

**Excitement:**

3: Ambivalent: It has merits (e.g., it reports state-of-the-art results, the idea is nice), but there are key weaknesses (e.g., it describes incremental work), and it can significantly benefit from another round of revision. However, I won't object to accepting it if my co-reviewers champion it.

**Paper Topic And Main Contributions:**

This paper captures the hallucination aspect arising from the presence of irrelevant documents and the generated overconfident answer scores when exploiting an LLM as a reader.

To reduce this problem, they define a two-step answering pipeline consisting of answer candidate generation and final answer selection.

1. they provide models with an unanswerable instruction, allowing them to refuse response to irrelevant documents.
    "If you don’t know the answer, return unanswerable"

2. adjust the answer scores by multiplying the query-document relevance.

**Questions For The Authors:**

1. What’s the difference between hallucination and incorrect answers in your work? Since you just use exact match (EM) accuracy as metric.

2. In Table 2, why the EM results for FLAN-T5-XL w/ HAS is different from the EM results in Table 1?

**Reasons To Accept:**

1. They empirically verify the efficacy of this simple approach in mitigating hallucination. The method achieves an increase in EM of 64%.

2. Conduct sound experiments to analyze including the impact of model size, and Answer Candidate Set.


**Reasons To Reject:**

1. Just mainly conduct experiments on FLAN-T5-XL (3B) and OPT-IML-MAX (1.3B). It’s better to explore larger models since your topic is about LLM.

**Reproducibility:**

4: Could mostly reproduce the results, but there may be some variation because of sample variance or minor variations in their interpretation of the protocol or method.

**Reviewer Confidence:**

3: Pretty sure, but there's a chance I missed something. Although I have a good feel for this area in general, I did not carefully check the paper's details, e.g., the math, experimental design, or novelty.

---

> ### Author Rebuttal · Authors · 2023-08-29
>
> We thank you for your thoughtful and valuable feedback on our work in the best regard. The positions of the graphs in Figure 5 have been swapped. We apologize for your confusion and it can be easily revised. In this response, we have restructured your comments into three distinct sections.
>
> ___
> ### **1. Feasibility of HAS on Much Larger Models (Reason to Reject)**
>
> We validated multiple models with fewer than 10 billion parameters within the limit of available academic resources. Our findings include the results obtained from FLAN-T5-XXL, which has 11 billion parameters, presented in Figure 3. The table below elucidates these outcomes, highlighting a consistent trend with Figure 2 across models exceeding the 10 billion parameters.
>
>
> |  DPR  | 10 Docs | 50 Docs | 100 Docs |
> |:-----------:|:-------:|:-------:|:--------:|
> | FLAN-T5-XXL | 29.7 | 23.0 | 19.9 |
> |  w/ HAS | 36.9 | 37.8 | 37.7 |
>
> The issue of hallucination defined previously is evident in all instances of utilizing LLMs [1,2]. Our scope does not fully encompass black-box models like GPT-3, which are constrained in obtaining token log probabilities. We mentioned this in the Limitation section [lines 303~307] as we acknowledge the same perspective of the review. We consider a universal approach encompassing diverse model types as one of our future research directions.
>
> ___
> ### **2. Relation between Hallucination and Exact Match (EM) (Q1)**
>
> Our research is motivated by hallucination [lines 51-65] and its impact is on the accuracy degradation of ODQA when augmenting the retrieved documents to LLMs.
>
> We find that LLMs augmented with retrieved sets confront hallucination on ODQA as caused by irrelevant documents on the input side and overconfident scores on the output side [lines 148-164]. These two facets of hallucination culminate in the eventual selection of an erroneous answer for the given question, thereby substantiating the observed decline in accuracy for ODQA tasks, as shown in Table 1 and Figure 2.
>
> Our analysis of these phenomena, as illustrated in Tables 3 and 5 and Figure 5, demonstrates the relevant mechanics. The findings in Tables 3 and 5 highlight that the existence of irrelevant documents within retrieved sets frequently prompts LLMs to generate inaccurate outputs, consequently resulting in a significant proportion of incorrect answers within the set of candidates. Meanwhile, Figure 5 visually represents the phenomenon of overconfidence in incorrect answers, which exhibits a distribution comparable to that of accurate answers. The coexistence of these two phenomena makes selecting the correct answer as the final output by LLMs challenging, hindering the advantage of retriever augmentation.
>
> ___
> ### **3. About Table 2 (Q2)**
>
> The difference between the exact match (EM) scores comes from the data division. The results in Tables 1 and 2 were based on the validation and test split, respectively. Because the validation split annotated the relevant documents for each query, the subsequent analyses are possible based on the results in Table 1, whereas Table 2 used the test split for a fair comparison with other supervised models. In the revision, we will provide a clear explanation of the distinction between the two results.
> ___
> #### **References**
>
> [1] SHI, Freda, et al. Large language models can be easily distracted by irrelevant context. In: International Conference on Machine Learning. PMLR, 2023. p. 31210-31227.
>
> [2] SI, Chenglei, et al. Prompting gpt-3 to be reliable. ICLR, 2023.

---

### Official Review · Reviewer_ubV7 · 2023-07-30

**Soundness:** 4

**Excitement:**

2: Mediocre: This paper makes marginal contributions (vs non-contemporaneous work), so I would rather not see it in the conference.

**Paper Topic And Main Contributions:**

In this work, the author(s) introduce Hallucination-aware Answer Selection (HAS), as part of a two-stage pipeline to mitigate the effects of LLMs hallucinating incorrect answers in the presence of irrelevant documents. To mitigate the challenge of irrelevant documents provided by the retriever model, the author(s) propose the use of an "unanswerable" instruction that prompts the language model to abstain from answering if the context does not provide enough information. Such contexts are subsequently removed from final answer selection. In the second stage, the author(s) propose to tackle the challenge of overconfident answer predictions from LLMs by regulating the score with the probability of generating the query given a document. Intuitively, if the document is irrelevant, the probability of the query would be low and hence the model down weights the score assigned to such an answer. Applying the two-stage pipeline on two instruction-tuned models, FLAN-T5 and OPT-IML-MAX, the proposed HAS method shows significant improvements over the vanilla models. Further analysis also indicates that the model is more robust to the addition of irrelevant documents, and works at any scale across multiple datasets.

**Questions For The Authors:**

- lines 207-210 mention that Sanh et al. 2022 also have an unanswerable instruction in their prompt. If so, what part of the unanswerability forms the novelty of this work? If not, could the author(s) kindly elaborate on the differences between the prior work and the proposal in this draft?
- Why does the absolute number of correct answers decrease after applying HAS to the FLAN-T5 (or OPTI-IML-MAX)? Does this imply the model is susceptible to removing some correct documents? Is there a pattern for when the correct documents are removed?

**Reference**:

Sanh et al. 2022: Multi-task prompted training enables zero-shot task generalization. ICLR 2022

**Reasons To Accept:**

- Performance increases as the number of retrieved documents increases in the model, implying that the model is indeed more robust to adding more irrelevant documents.
- Experiments with multiple models and across multiple datasets highlight the generality of the proposed method.

**Reasons To Reject:**

- **Novelty and Clarity**: Inferring how the abstention section works is hard when the description is restricted to just four lines (L175-179). It is unclear what answers form a part of the unanswerable set. Is it just a single answer "unanswerable"? Or is there some other mechanism used? Further, by the authors' own admission, the technique of abstention and score regulation has also been explored in prior works in the question-answering setting. This calls to question the novelty of the proposed method beyond the fact that it has been applied to zero-shot models.


**Reproducibility:**

4: Could mostly reproduce the results, but there may be some variation because of sample variance or minor variations in their interpretation of the protocol or method.

**Reviewer Confidence:**

3: Pretty sure, but there's a chance I missed something. Although I have a good feel for this area in general, I did not carefully check the paper's details, e.g., the math, experimental design, or novelty.

**Typos Grammar Style And Presentation Improvements:**

- line 221: improvement of 231% → improvement of 231% on SQUAD.
- line 262: kindly mention that this is the percentage improvement in the number of correct answers for FLAN-T5 and OPT-IML-MAX respectively.

---

> ### Author Rebuttal · Authors · 2023-08-29
>
> Thank you for your time and effort in reviewing our paper and providing valuable comments. The subsequent contents pertain to the response to your comments and explanation of our work, including additional results. The positions of the graphs in Figure 5 have been swapped. We apologize for your confusion and it can be easily revised. The responses will be reflected in the revision of our work. We reorganized your comments into three sections for clarity.
>
> ___
> ### **1. Clear Description of Our Proposed Method, HAS (Reason to Reject)**
>
> We explained the two-step mechanism of HAS in Section 3.3 with a concise overview [lines 166-174] and in Appendices B.3 and B.4. We will revise the description of the method for a clearer understanding, including the contents of the Appendix.
>
>  HAS consists of two stages: document and answer selection. The document selection is for not responding to the irrelevant documents in the retrieved sets. By adding the unanswerable instruction, *the negation-based prompting*, to the original instruction [line 205-209], documents and answers from the answer candidate set $U$ are filtered out when their outputs are *“unanswerable”* or *“Answer not in context”*. Subsequently, the answer selection alleviates the overconfidence of generation probability by reflecting the relevance between the question and the document [lines 181-186]. The answer, having the highest value from Equation (2) and the query generation score described in Appendix B.4, is chosen as the final answer to the question.
>
> For example, let the relevant context $d$ and the irrelevant documents $d_{n1}$ and $d_{n2}$ in the retrieved document set $D$ for the given query $q$. For the document selection stage, if LLM generates $a_{n1}$ as *“unanswerable”* from the $d_{n1}$, the pair $(a_{n1},d_{n1})$ is excluded to the answer candidate set $A$. Then, the answer candidate set $A$ is $\lbrace(a_{n2},d_{n2}), (a,d)\rbrace$. For the answer selection stage, we calculate the answer score for each pair based on Equation (2). Considering the relevance of context, the score of $(a,d)$ is higher than the score of $(a_{n2}, d_{n2})$.
>
> ___
> ### **2. The Novelty of HAS & Differences from Previous Approaches (Reason to Reject & Q1)**
>
> Each component of the HAS framework brings novel insights to questionable and underexplored domains, as described in Appendix A. Our novelty comes from bridging this gap by empirically demonstrating both negation-based prompting & score regulation and effective mitigation of hallucination induced by both input and output.
>
>
> Although the integration of unanswerable instructions within the pre-training of LLMs has been acknowledged [1], it remains one of the instructions describing diverse tasks aimed at enhancing generality, often underestimating its potential for negation. [2,3] also pointed out the negative impact of irrelevant or hallucinated contexts when exploiting LLMs for diverse tasks, underscoring the necessity of negation.  To the best of our knowledge, the score regulation based on query generation is first utilized for the alleviation of overconfidence in LLMs, while its utilization is usually on document ranking.
>
>
> ___
> ### **3. Distribution Shift of Answer Candidate Set $U$ (Q2)**
>
> HAS excludes irrelevant documents as much as possible and utilizes only answers from relevant documents. Negating irrelevant contexts involves a trade-off with the number of accurate responses, but we confirmed through Table 1 and Figure 1 that this gives rise to positive results on ODQA.
>
> For further analysis, Table 5 within the Appendix highlights that most of the incorrect answers are from irrelevant contexts, leading to a significant proportion of answer candidate set $U$. HAS has efficiently diminished the count of incorrect answers originating from irrelevant contexts while preserving the number of accurate answers from relevant contexts.
>
>
>
> ___
> #### **References**
>
> [1] Sanh et al. 2022: Multi-task prompted training enables zero-shot task generalization. ICLR 2022
>
> [2] SHI, Freda, et al. Large language models can be easily distracted by irrelevant context. In: International Conference on Machine Learning. PMLR, 2023. p. 31210-31227.
>
> [3] SI, Chenglei, et al. Prompting gpt-3 to be reliable. ICLR, 2023.

---

### Official Review · Reviewer_CcX8 · 2023-08-05

**Soundness:** 4

**Excitement:**

2: Mediocre: This paper makes marginal contributions (vs non-contemporaneous work), so I would rather not see it in the conference.

**Paper Topic And Main Contributions:**

This paper proposes to use LLMs as the reader module in the traditional retrieve-read framework for open-domain question answering, under a zero-shot setting. Compared to previous small-sized supervised readers, the zero-shot approach eliminates the need for downstream task training and annotated query-document datasets. However, using LLMs also introduces hallucination issues, exacerbated when retriever results contain irrelevant documents. To address this, the authors propose Hallucination-aware Answer Selection (HAS) to alleviate the impact of irrelevant documents and overconfident LLMs, which is to prompt the LLMs to filter out the irrelevant documents via unanswerable instruction. Experiments show that HAS achieves average EM improvements of 67% and up to 231%. However, performance does not significantly exceed previous fine-tuning methods like DPR and FID.

**Questions For The Authors:**

See "reasons to reject"

**Reasons To Accept:**

1.	The paper proposes using LLMs as readers in retrieve-read with a simple and effective HAS to handle irrelevant information and hallucination caused by inaccurate retrievers. Despite no significant gains over previous methods, the zero-shot setting greatly reduces training costs and annotation needs.
2.	Experiments validate the efficacy of HAS, showing clear improvements over not using HAS across datasets and retrievers.

**Reasons To Reject:**

1.	HAS adds document and answer selection modules using LLMs to filter retriever results. The increased inference costs were not considered. With a large number of inference, LLM costs may exceed training a small model. HAS does not outperform small supervised model solutions, sometimes underperforming. Given LLM abilities to select relevant documents, ideas like COT prompt design could be explored to have LLMs filter then answer to achieve HAS effects. In this case, what is the advantage of employing LLMs in this task?
2.	More popular LLAMA and ChatGPT models were not tested. The smaller FLAN-T5 and OPT-IML have less zero-shot capacity than current best open LLM models, so HAS may perform better on stronger models.
3.	It may be unnecessary to follow the traditional retrieve-read framework for LLMs in open-domain QA. LLMs could ingest open-domain data during pretraining, supplement with lang-chain, and benchmark current LLM open-domain performance.
4.	I think the title is somehow misleading. Hallucination is indeed an issue for LLMs. However, this paper is not to alleviate hallucination in LLMs, it tries to filter out the irrelevant documentations during retrieval procedure for open-domain QA tasks, which is not to improve the LLMs themselves. Also this brings limitation to the study.

**Reproducibility:**

3: Could reproduce the results with some difficulty. The settings of parameters are underspecified or subjectively determined; the training/evaluation data are not widely available.

**Reviewer Confidence:**

3: Pretty sure, but there's a chance I missed something. Although I have a good feel for this area in general, I did not carefully check the paper's details, e.g., the math, experimental design, or novelty.

---

> ### Author Rebuttal · Authors · 2023-08-29
>
> We appreciate your thoughtful and helpful comments to increase the quality of our manuscript. The following discussion refers to our submitted manuscript as our paper. Additional explanations and materials are for enhanced clarity, which will be reflected in the revised version. The positions of the graphs in Figure 5 have been swapped. We apologize for your confusion but it can be easily fixed.
> ___
> ### **1. Hallucination Induced by Easily Distracted and Overconfident LM (Q4)**
>
> First, we want to point out a misunderstanding of our research motivation, the hallucination of LLM, as mentioned in lines 51-65.
>
> Hallucination generally refers to the generation of plausible yet untrue text. Its reasons are diverse on specific tasks but usually inherent in the mechanism of LLM [lines 119-130]. Previous work on open-domain question answering (ODQA) points out that LLMs are easily distracted to produce incorrect answers by the irrelevant context from retrieved sets [1,2,3] and their overconfidence (i.e., high log probability of sequences) from the decoding strategy [4,5,6]. Lines 51-59 are supported by studies about the impact of irrelevant context leading to accuracy degradation. Figure 5 indicates that the overconfidence of LLMs on output makes it hard to identify the correct ones between generated answers.
>
> In addition, we empirically confirm that both causes from input and output are related through our analysis of Tables 3, 5, and Figure 5 of our paper. Table 3 illustrates a high frequency of incorrect answers among the answer candidates, and Table 5 specifies that these errors stem from irrelevant documents. Then, Figure 5 describes overconfidence in answer scores on a default setting, which means that LLMs have trouble identifying correct results. Our proposed method, HAS, successfully alleviates such problems depicted in all analyses.
>
> |                    | NQ | TQA | SQuAD |
> |:--------------------------------------:|:-----:|:-----:|:-----:|
> |  FLAN-T5-XL w/o Context (Lower Bound) | 12.09 | 25.30 | 5.86 |
> |  FLAN-T5-XL w/ Top-100 Docs    | 15.90 | 39.17 | 10.30 |
> |  FLAN-T5-XL w/ HAS        | 37.96 | 68.22 | 34.12 |
> | FLAN-T5-XL w/ Gold Document (Upper Bound) | 54.53 | 75.27 | 73.84 |
>
> For clear understanding, the table above shows that naïvely augmenting the retrieved document enhances the performance marginally, leaving much room for improvement. In terms of utilizing LLMs, this underscores the impacts of hallucination rather than augmenting with external knowledge. Ours is much more aligned with the upper bound.
>
> This highlights that *ours effectively filtered out the negative impact of such a context, resulting in alleviating hallucination in LLMs*.
>
> ___
> ### **2. Necessity of Incorporating LLM into the Traditional Retrieve-then-Read Pipeline (Q1 & Q3)**
>
> We categorized this section into two research questions mentioned in the reasons to reject.
>
> **1) LLMs themselves have impressive abilities in answering questions requesting background knowledge.**
>
> Although the open-domain data is included in the training datasets, LLMs are still challenged under the closed-book setting for knowledge-intensive tasks. Our work is aligned with the concurrent studies that retrieval augmentation provides a promising path toward improving performance [7,8,9]. Furthermore, the langchain already incorporates the retriever modules to locate evidence information, highlighting the significance of retriever augmentation for LLMs [10]. Our proposed method, HAS, is suggested to utilize the retrieve-then-read framework in handling the hallucination mentioned above and eliciting the full performance of LLMs.
>
> **2) LLMs with the proposed methods do not outperform small supervised model solutions, considering their increased inference cost.**
>
> Exploiting LLMs brought us to the expectation of a zero-shot approach and generality of LLMs [lines 37-50, 117-118]. Regarding perspective, comparing a supervised model directly with LLMs is not straightforward, given the substantial training costs and the need for ongoing updates to maintain the performance on out-domain datasets.
>
> We conducted additional experiments on out-domain datasets with the instruction-tuned model, FLAN-T5-base, and large, which have the same size as FiD readers. These experiments are conducted on the supplementary dataset, RealTimeQA [11], based on current world knowledge.
>
>
> |     | Param | Train Set | SQuAD | RealTimeQA | Avg. |
> |:-----------------:|:-----:|:---------:|:-----:|:----------:|:----:|
> |  FiD - Base  | 250M |  NQ  | 34.1 |  29.8  | 32.0 |
> |         |   |  TQA  | 27.5 |  33.2  | 30.4 |
> |  FLAN-T5-Base |   |  X  | 10.5 |  20.9  | 15.7 |
> |   w/ HAS   |   |     | 36.4 |  31.6  | 34.0 |
> |  FiD - Large  | 780M |  NQ  | 36.2 |  34.0  | 35.1 |
> |         |   |  TQA  | 31.5 |  34.4  | 33.0 |
> | FLAN-T5-Large |   |  X  | 10.8 |  20.1  | 15.5 |
> |   w/ HAS   |   |     | 41.5 |  34.7  | 38.1 |
> | FLAN-T5-XL w/ HAS | 3B |  X  | 43.5 |  35.8  | 39.7 |
>
>
> The results follow a tendency shown in Table 2 of the paper with the FLAN-T5-XL. The small supervised models perform well over in-domain datasets (i.e., train datasets), but their performance does not remain consistent over out-domain datasets such as SQuAD and RealTimeQA [lines 249-254]. Using LLMs alone does not enhance performance. Notably, even with the smaller scale of the instruction-tuned model, our proposed approach is found to harness its capacity for generalization in ODQA.
>
> ___
> ### **3. Other Possible Prompt Designs such as Chain-of-Thoughts (CoT) (Q1)**
>
> We wish to clarify that CoT and our proposed method, HAS, take a different approach. While CoT has gained attention for simply mimicking the reasoning process of a human with performance improvement, it seems inappropriate to solve the problems we described [3,12,13]. In particular, [3] demonstrates that CoT cannot alleviate the negative impact of the irrelevant context even in arithmetic reasoning. The following table proves that the CoT design cannot solve the problem induced by hallucination when top-100 documents from DPR are transmitted to FLAN-T5-XL.
>
>
> |   | NQ-Dev | TQA-Dev |
> |:------------:|:------:|:-------:|
> | FLAN-T5-XL | 15.90 | 39.17 |
> |  w/ CoT  | 13.00 | 34.52 |
> |  w/ HAS  | 37.96 | 68.22 |
> | w/ CoT & HAS | 37.04 | 65.21 |
>
> Compared to this, our proposed method, HAS, is a *negation-based prompting* that lets the model not answer when models are unsure. To the best of our knowledge, we have empirically validated the effectiveness of such a prompt design on negation. In contrast, the current prompt design is targeted at generating self-supervised evidence.
>
> ### **4. Validation on Much Larger Models (Q2)**
>
> We validate multiple models under 10 billion parameters available at our academic resources. Also, please note that we showed the result of FLAN-T5-XXL having 11 billion parameters in Figure 3. The table below gives detailed results showing a similar tendency on even over 10B models for clear understanding.
>
> |      | 10 Docs | 50 Docs | 100 Docs |
> |:-----------:|:-------:|:-------:|:--------:|
> | FLAN-T5-XXL | 29.7 | 23.0 | 19.9 |
> |  w/ HAS | 36.9 | 37.8 | 37.7 |
>
> The problem of hallucination, as previously defined, is apparent in all cases involving models over 100B parameters [3, 8]. However, it is essential to emphasize that our focus does not extend to black-box models such as GPT-3, primarily due to our constrained access to token log probabilities. This limitation was articulated in the Limitation section in lines 303~ 307, aligned with the viewpoint expressed in the previous work [3,8].
>
> ___
> #### **References**
>
> [1] LI, Daliang, et al. Large language models with controllable working memory. In: Findings of the Association for Computational Linguistics. 2023. p. 1774-1793.
>
> [2] SU, Dan, et al. Read before generate! faithful long form question answering with machine reading. In: Findings of the Association for Computational Linguistics. 2023. p. 744-756.
>
> [3] SHI, Freda, et al. Large language models can be easily distracted by irrelevant context. In: International Conference on Machine Learning. PMLR, 2023. p. 31210-31227.
>
> [4] LIN, Stephanie; HILTON, Jacob; EVANS, Owain. Teaching models to express their uncertainty in words. Transactions on Machine Learning Research, 2022.
>
> [5] MIELKE, Sabrina J., et al. Reducing conversational agents’ overconfidence through linguistic calibration. Transactions of the Association for Computational Linguistics, 2022, 10: 857-872.
>
> [6] XIONG, Miao, et al. Can LLMs Express Their Uncertainty? An Empirical Evaluation of Confidence Elicitation in LLMs. arXiv preprint arXiv:2306.13063, 2023.
>
> [7] MALLEN, Alex, et al. When not to trust language models: Investigating effectiveness of parametric and non-parametric memories. In: Proceedings of the 61st Annual Meeting of the Association for Computational Linguistics (Volume 1: Long Papers). 2023. p. 9802-9822.
>
> [8] SI, Chenglei, et al. Prompting gpt-3 to be reliable. In: International Conference on Learning Representation. 2023.
>
> [9] KANDPAL, Nikhil, et al. Large language models struggle to learn long-tail knowledge. In: International Conference on Machine Learning. PMLR, 2023. p. 15696-15707.
>
> [10] LangChain AI. 2023. Available online: https://python.langchain.com/docs/modules/data_connection/
>
> [11] KASAI, Jungo, et al. RealTime QA: What's the Answer Right Now?. arXiv preprint arXiv:2207.13332, 2022.
>
> [12] LI, Junlong; ZHANG, Zhuosheng; ZHAO, Hai. Self-prompting large language models for open-domain qa. arXiv preprint arXiv:2212.08635, 2022.
>
> [13] TURPIN, Miles, et al. Language Models Don't Always Say What They Think: Unfaithful Explanations in Chain-of-Thought Prompting. arXiv preprint arXiv:2305.04388, 2023.

---

### Meta-Review · Area_Chair_A6FV · 2023-09-23

**Recommendation:** 3

**Metareview:**

The paper proposed a method to improve results in open-domain QA. Typically, open-domain QA is done by 1) retrieving relevant docs, 2) running a "reader" that reads the question + the retrieved documents and outputs the answer. The proposed method focuses on filtering out noisy retrieved documents. It allows the reader to output "no answer" if the retrieved document is not useful.

The model is implemented using FLAN-T5-XL-3B and using OPT-IML-MAX-1.3B. Results show performance gains on NQ, TQA, WebQ, SQuAD.

Reviewers found the result to be generally sound.
Some reviewers complained about the choice of the base models, which I don't think is a fair criticism; 3B is a reasonable size + not every researcher can run the largest available model.
Others commented on the novelty of the work, and yes, the idea of giving the model the ability to say "I don't know" is not new. The details of how it is implemented are a little different from prior work, though.


Nit: I agree with R1 that using the word hallucination in this context is inaccurate. Hallucination usually means generating output not supported by the input. Here, the output is supported by one of the retrieved documents, it just happened that it was the wrong document.

---

### Decision · Program_Chairs · 2023-10-07

**Decision:**

Accept-Findings

**Comment:**

The paper proposed a method to improve results in open-domain QA. Typically, open-domain QA is done by 1) retrieving relevant docs, 2) running a "reader" that reads the question + the retrieved documents and outputs the answer. The proposed method focuses on filtering out noisy retrieved documents. It allows the reader to output "no answer" if the retrieved document is not useful.

The model is implemented using FLAN-T5-XL-3B and using OPT-IML-MAX-1.3B. Results show performance gains on NQ, TQA, WebQ, SQuAD.

Reviewers found the result to be generally sound.
Some reviewers complained about the choice of the base models, which I don't think is a fair criticism; 3B is a reasonable size + not every researcher can run the largest available model.
Others commented on the novelty of the work, and yes, the idea of giving the model the ability to say "I don't know" is not new. The details of how it is implemented are a little different from prior work, though.


Nit: I agree with R1 that using the word hallucination in this context is inaccurate. Hallucination usually means generating output not supported by the input. Here, the output is supported by one of the retrieved documents, it just happened that it was the wrong document.